# Emerging Roles of β-Glucanases in Plant Development and Adaptative Responses

**DOI:** 10.3390/plants11091119

**Published:** 2022-04-20

**Authors:** Thomas Perrot, Markus Pauly, Vicente Ramírez

**Affiliations:** Institute for Plant Cell Biology and Biotechnology—Cluster of Excellence on Plant Sciences, Heinrich Heine University Düsseldorf, 40225 Düsseldorf, Germany; thomas.perrot@hhu.de (T.P.); m.pauly@hhu.de (M.P.)

**Keywords:** β-glucanases, β-glucans, cell wall polysaccharides, plant development, environmental stress

## Abstract

Plant β-glucanases are enzymes involved in the synthesis, remodelling and turnover of cell wall components during multiple physiological processes. Based on the type of the glycoside bond they cleave, plant β-glucanases have been grouped into three categories: (i) β-1,4-glucanases degrade cellulose and other polysaccharides containing 1,4-glycosidic bonds to remodel and disassemble the wall during cell growth. (ii) β-1,3-glucanases are responsible for the mobilization of callose, governing the symplastic trafficking through plasmodesmata. (iii) β-1,3-1,4-glucanases degrade mixed linkage glucan, a transient wall polysaccharide found in cereals, which is broken down to obtain energy during rapid seedling growth. In addition to their roles in the turnover of self-glucan structures, plant β-glucanases are crucial in regulating the outcome in symbiotic and hostile plant–microbe interactions by degrading non-self glucan structures. Plants use these enzymes to hydrolyse β-glucans found in the walls of microbes, not only by contributing to a local antimicrobial defence barrier, but also by generating signalling glucans triggering the activation of global responses. As a counterpart, microbes developed strategies to hijack plant β-glucanases to their advantage to successfully colonize plant tissues. This review outlines our current understanding on plant β-glucanases, with a particular focus on the latest advances on their roles in adaptative responses.

## 1. Introduction

The plant cell wall is a dynamic composite structure with diverse functions including mechanical and structural support for plant growth, compartmentalization of specialized cells, and the integration of cell-to-cell communication and interactions with the environment [1,2]. Given these essential roles, plants have evolved intricate mechanisms to assemble, modify, and dismantle the diverse wall components. β-glucans including cellulose, callose, xyloglucan, and mixed-linkage glucan (MLG) are prevalent wall structural constituents in most plant cell types. These polysaccharides share a backbone composed of d-glucopyranosyl building blocks linked by β-1,4 and/or β-1,3 bonds, which can contain additional sidechain substitutions. The abundance, length, and associations of these glucan structures are modified during development and in response to environmental cues [3,4]. The diversification of glucan structures found in plant walls has been accompanied by the coevolution of specific hydrolases allowing for their modification/degradation. Plant genomes encode diverse types of enzymes able to hydrolyse β-glucans. Endo-β-glucanases, the most abundant type in plants, randomly cleave internal β-d-glucosidic linkages in the glucan backbone, while exo-β-glucanases act processively on both ends of the glucan chain, releasing oligosaccharides. Finally, β-glucosidases hydrolyse terminal β-d-glucosyl residues releasing β-d-glucose. These enzymes can be further classified in many ways depending on the reaction mechanism used, the chemical reaction they catalyse, or amino acid sequence-related aspects. Based on common amino acid sequences and protein fold structures, β-glucanases are typically distributed in multiple glycoside hydrolase (GH) families according to the classification of Carbohydrate-Active enZymes (CAZy) [5]. Frequently, Enzyme Commission (EC) number(s) are also used to designate the biochemical reaction(s) catalysed by these proteins [6]. Depending on the glycoside bond they hydrolyse, three main types of β-glucanases can be found in plants: β-1,4-glucanases, β-1,3-glucanases and β-1,3-1,4-glucanases. 

1,4-β-glucanases are enzymes able to hydrolyse the 1,4-glycosidic bond between two contiguous d-glucopyranose units (Figure 1). This bond is found in the structure of cellulose, the most abundant polysaccharide in plant walls. 1,4-β-glucanases, initially termed “cellulases”, include several types of enzyme activities needed to degrade cellulose, such as endoglucanases (EC 3.2.1.4), cellobiohydrolases (EC 3.2.1.91), and glucosidases (EC 3.2.1.21) [7,8,9]. Plant cellulases are classified in the GH9 family [5,10]. Although most of them exhibit only limited activity on crystalline cellulose, they are able to hydrolyse amorphous regions of cellulose and soluble cellulose derivatives such as carboxymethyl cellulose. There is some evidence suggesting that these enzymes are also able to cleave other non-cellulosic polysaccharides containing contiguous (1,4)-β-glucosyl residues in their backbone, including MLG, xyloglucan, and glucomannan, although characterization of the hydrolytic activities on different substrates has been limited to a handful of plant GH9 proteins [11]. GH9 β-1,4-glucanases have been implicated in several aspects of cell wall metabolism in higher plants, including cellulose biosynthesis and degradation, modification of the association of cellulose microfibrils with other wall polysaccharides, or wall loosening during cell elongation [12,13,14]. 

The term β-1,4-glucanase includes additional enzyme activities such as those involved in the hydrolysis of xyloglucans, heavily xylose-substituted 1,4-β-glucans [15]. Xyloglucan endohydrolases (XEH; EC 3.2.1.151) cleave the xyloglucan chain, releasing oligosaccharides. However, some of these enzymes also exhibit transglucosylase activity (XET; EC 2.4.1.207) being able to covalently link these oligosaccharides onto the non-reducing terminal end of the glucose moiety of other xyloglucan and cellulose acceptors [16,17,18,19]. This group of enzymes, generally termed xyloglucan endotransglucosylase/hydrolases (XTH) has important roles in wall polymer rearrangement, polymer integration into the wall, and cell expansion, and has also been implicated in plant responses towards abiotic and biotic stresses. Several reviews have recently addressed this transglycosylase class of plant β-1,4-glucanases [20,21], and they will not be further considered in this review.

β-1,3-glucanases catalyse the hydrolysis of glucans containing contiguous β-1,3-linked glucosyl residues (Figure 1; [22]). β-1,3-glucanases are responsible for the degradation of callose in plants, but they can also hydrolyse the β-1,3- and β-1,3-1,6-glucans found in the walls of intruding fungi [23,24]. They are also called laminarinases, as they are able to cleave laminarin, a linear β-1,3-glucan displaying occasional β-1,6-branches found in brown algae [25]. Laminarin and the structurally similar β-1,3-glucans paramylon and pachyman are usually employed to characterize in vitro activities of those enzymes. Plant β-1,3-glucanases are classified together with β-1,3-1,4-glucanases in the GH17 family of glycosyl hydrolases [26].

β-1,3-1,4-glucanases only hydrolyse β-1,4-glucosidic linkages if an adjacent β-1,3-glucosyl linkage is present towards the non-reducing end of the substrate (Figure 1). These enzymes are highly specific, and they are not able to hydrolyse β-1,3- or β-1,4-glucans [8,27,28]. In plants, these enzymes are involved in the degradation of MLG, a hemicellulosic polysaccharide prevalent in the walls of grass species. β-1,3-1,4-glucanases are also known as lichenases or licheninases, named after their activity on lichenin (moss starch), a complex glucan occurring in Parmeliaceae lichens [29,30]. This β-1,3-1,4-glucan differs from those characterized in grasses in having a much higher proportion of cellotriosyl to cellotetraosyl units [31]. In addition, despite having similar substrate specificities, plant β-1,3-1,4-glucanases have quite distinct amino acid sequences and 3D structures compared with microbial licheninases; thus, the term β-1,3-1,4-glucanase will be used [8,32,33].

This review will encompass the evolutionary relationships, classification, and biological roles proposed for the following three types of plant endo-β-glucanases (referred as β-glucanases from here): β-1,4-glucanase (“cellulase” EC 3.2.1.4), β-1,3-glucanase (EC 3.2.1.39) and β-1,3-1,4-glucanase (EC 3.2.1.73). The specificities of these enzymes on their biological β-glucan substrates are shown in Figure 1.

## 2. β-1,4-Glucanases

### 2.1. Classification and Evolutionary Origin

Enzymes able to use cellulose as substrate are found in many prokaryotic and eukaryotic organisms, including bacteria, fungi, nematodes, animals and plants [34]. However, the hydrolytic mechanisms used and thus the protein domains involved are different. In plants, a similar number of GH9 β-1,4-glucanases (EC 3.2.1.4) are found in diverse species such as Arabidopsis (25), poplar (25) barley (22), rice (24) sorghum (23) or Brachypodium (23) [34,35,36,37]. All these plant β-1,4-glucanases harbour a catalytic domain containing two conserved motifs required for the activity distinguishing GH9 from other hydrolases [9,38]. In the first motif, a histidine improves the stability of the substrate in the active site, while in the second motif, an aspartate (the nucleophile base) and a glutamate (the proton donor) are required for the cleavage of the glycosidic bond [39]. The presence of additional domains allows one to classify plant GH9 β-1,4-glucanases into three distinct structural subclasses: GH9A, GH9B and GH9C (Figure 2; [30,31,32]). In general, the domain structures and genomic intron–exon organization seem to be conserved in GH9 subgroups from distantly related plants such as monocots and dicots [36]. In the GH9A subgroup, the catalytic domain is preceded by a cytosolic N-terminal extension followed by a membrane-spanning domain that anchors the protein to the plasma membrane and/or to intracellular organelles. In subgroups GH9B and GH9C, the catalytic domain is instead preceded by a signal peptide for ER targeting and secretion. Subgroup GH9C members possess an additional long carboxyl-terminal extension containing a carbohydrate binding module (CBM49), which can bind cellulose in vitro (Figure 2; [36]). 

Structural modelling and phylogenetic studies comparing the GH9 family in plant and non-plant taxa suggest that classes A and B of modern vascular land plants may have emerged by diverging directly from a common GH9C ancestor [42]. Although evidence is mostly based on in vitro activities and 3D modelling and should thus be interpreted with caution, some relationships between the three domain structures and biological functions have been proposed. The CBM49 domain, exclusive of plant GH9C β-1,4-glucanases, would confer the ability to digest crystalline substrates in a manner reminiscent from ancestral non-plant taxa such as bacteria. Subsequent loss of CBM49 during plant evolution and replacement of the signal peptide by a transmembrane region would have resulted in the appearance of the B and A classes, respectively. These events would have resulted in the development of new cellulose editing/modifying functions, where A and B classes would only be suited to digest amorphous regions of cellulose [42,43]. 

### 2.2. Biological Roles

Membrane-bound GH9A β-1,4-glucanases, lacking a CBM49 domain, would participate primarily in the assembly and repair/editing of cellulose microfibrils during cellulose biosynthesis in plants. Accordingly, KORRIGAN (KOR) GH9A β-1,4-glucanases are anchored to the plasma membrane associated with the cellulose synthase complex (CSC) and are required for the correct cellulose biosynthesis in both primary and secondary walls [14]. Several KOR genes have been identified in diverse plant species, KOR1 being the best studied member of this group [12,44]. In vitro, KOR1 is able to hydrolyse amorphous but not crystalline cellulose [45,46]. KOR1 β-1,4-glucanase activity is required for the efficient function of the CSC, and kor1 mutations result in CSC intracellular trafficking defects, reduced cellulose polymerization velocity, and aberrant crystalline cellulose deposition [13,14,47,48,49,50]. The precise mechanism is unclear, but it has been hypothesized that KOR1 would hydrolyse disordered amorphous glucan chains during cellulose synthesis. This “proofreading” activity might prevent unwanted interactions of microfibrils from adjacent CSCs, ensuring the correct fibril assembly. A second hypothesis is that KOR1 might cleave cellulose chains to terminate their synthesis once an adequate size is reached and before the CSC is internalized from the plasma membrane for recycling through endocytosis. A third possibility states that KOR1 is involved in the synthesis of a priming molecule required for cellulose synthesis initiation [45,51,52]. KOR1 function in cellulose biosynthesis seems to be conserved in several plant species including rice (*Oryza sativa*), tomato (*Solanum lycopersicon*), or *Populus* sp. [53,54,55,56]. Other GH9A genes with different expression profiles are also involved in cellulose synthesis in specific tissues and developmental conditions. For example, *Populus* PtrCel9A6 and PtrGH9A7 are two membrane-anchored β-1,4-glucanases which seem to regulate cellulose synthesis during xylem differentiation and lateral root formation, respectively [57,58].

Some defence-related defects have been associated with GH9A misregulation, although given the importance of a correct cellulose deposition in plants, these phenotypes are likely an indirect consequence of secondary metabolic alterations [55,59]. For example, the kor1-1 Arabidopsis mutant exhibits increased disease resistance to *Pseudomonas syringae* associated with an enhanced production of jasmonic acid, antagonizing the activation of salicylic acid-mediated defences [59]. Such an altered defence status has been further linked to an enhanced mycorrhization in KOR1-downregulated Populus deltoides plants [55].

GH9B β-1,4-glucanases are secreted into the apoplast where they have been typically associated with the wall remodelling in muro. Due to the lack of CMB49, these enzymes have limited or no hydrolytic activity on crystalline cellulose (Figure 2). Instead, GH9B could hydrolyse amorphous regions of cellulose involved in the interaction with other wall polymers altering the mechanical properties and promoting wall loosening or disassembly required in physiological processes such as cell expansion, fruit maturation or leaf abscission [9,60,61,62,63,64,65]. Evidence of biochemical activity for this type of enzyme is scarce, and their biological substrate(s) are usually inferred from indirect evidence. For example, a fruit-specific expression pattern has been observed for several β-1,4-glucanases in tomato, strawberry, avocado or pepper [60,66,67,68]. Fruits entering the ripening phase stop growing and tissues start softening, which is linked to wall structural alterations. FaEG1, a secreted GH9B β-1,4-glucanase, is specifically induced during strawberry ripening (Figure 2; [69]). Based on compatible cellulose- and xyloglucan-FaEG1 molecular docking simulations, it was suggested that FaEG1 might function in the disassembly of the cellulose–hemicellulose fraction during the ripening of strawberry fruit [69].

In tomato, silencing of the SlGH9B2 1,4-glucanase results in an increase in the force required to detach fruits from the plant [61]. Together with the localized expression in the abscission zone, it was suggested that SlGH9B2 is involved in wall disassembly at specific sites of organ detachment. Similarly, Arabidopsis Cel3 and Cel5 GH9Bs are thought to promote wall loosening required for the detachment of root cap cells during root development and in response to biotic and abiotic rhizosphere stresses [63,64,65]. 

In poplar, it was hypothesized that Cel1 and Cel2 promote leaf growth by hydrolysing non-crystalline regions of cellulose and by releasing xyloglucans intercalated within the disordered domains of cellulose microfibrils. This would reduce wall stress, increasing the accessibility of wall components to additional enzymatic modifications required for wall restructuring during cell expansion and growth [62,70,71,72]. Accordingly, overexpression of the poplar Cel1 1,4-β-glucanase decreases the amount of xyloglucan cross-linked with cellulose microfibrils, increasing wall plasticity and promoting enlargement of plant cells [72]. Similarly, overexpression of OsGH9B1 and OsGH9B3 1,4-β-glucanases in rice results in reduced cellulose crystallinity and degree of polymerization. Although only slight differences in wall polymer composition, stem mechanical strength, and biomass yield were observed, GH9B1/3 overexpression enhanced wall enzymatic digestibility likely caused by an enhanced accessibility due to a more open wall architecture [73]. 

GH9B 1,4-β-glucanases have also been involved in the maintenance of secondary wall mechanical strength [74]. GH9B5 is highly expressed during secondary wall development in poplar and Arabidopsis. AtGH9B5 t-DNA knock-out mutant walls exhibit decreased mechanical strength accompanied by reduced xylose and glucose content in the hemicellulosic fraction, while overexpression of AtGH9B5 using a promoter specific to secondary wall development led to the opposite phenotype with an increase in cellulose crystallinity. According to these results, it was proposed that GH9B5 might facilitate the coating of cellulose microfibrils with xylans during secondary wall formation resulting in strengthened walls [74]. Unlike other GH9B members, GH9B5 contains a predicted GPI anchor, thus linking the GH initially to the plasma membrane [75]. Proteomic analyses are consistent with such an association with the plasma membrane, potentially bringing GH9B5 in close contact with the CSC (Figure 2; [74,76,77]). 

A new role for a selected clade of secreted GH9B β-1,4-glucanases has recently been proposed in the cell-to-cell adhesion process during grafting and parasitic plant–host interaction. Successful grafting requires the adhesion of facing cells at the graft boundary [78]. NbGH9B3 is induced together with other wall remodelling genes in these graft junctions. NbGH9B3 misregulation results in a significant decrease in the graft success percentage in multiple inter- and intra-specific grafts [79]. Although the exact substrate involved is still unknown, it has been proposed that NbGH9B3 might target amorphous cellulose regions at the graft boundary, promoting cell adhesion and the formation of xylem bridges required for a compatible graft [79]. The expression of similar GH9B β-1,4-glucanases during compatible grafting is conserved across other plant species such as soybean (*Glycine max*), morning glory (*Ipomoea nil*), or Arabidopsis, suggesting a common mechanism. 

Some parasitic plants seem to use a comparable mechanism to disrupt host cell wall barriers and develop similar xylem bridges, allowing the parasitic plant to colonize host tissues. During *Phtheirospermum japonicum*–*Arabidopsis* interaction, *P. japonicum* secretes the PjGH9B3 ortholog in the periphery of the haustorium in direct physical contact with the host tissue. As a result, the thickness of the host wall at the interface area between both parasite and host plant cells is decreased, and xylem bridges are formed, allowing a successful cell-to-cell adhesion and further colonization [80]. 

Alterations in the wall structure caused by GH9Bs have also been involved in the interaction between plants and microbial pathogens. Arabidopsis GH9B1 and GH9B2 and their respective tomato homologs are required for the establishment of balanced defence responses against pathogens with different lifestyles. Although the exact role is unclear, it has been speculated that a lack of SlGH9B1/2 might modify the structure and properties of the wall, altering its signalling abilities [81,82]. 

Expression of specific GH9B β-glucanases has also been associated with a more efficient defensive response against nematode attacks. According to the proposed mechanism, plants activate the expression of secreted β-glucanases at the nematode feeding site to halt the infection, likely interfering with pathogen development [83]. For example, AtGH9B Cel6 is highly expressed during compatible plant–nematode interactions, and ectopic expression of AtCel6 in soybean roots reduces disease susceptibility to soybean cyst nematode (*Heterodera glycines*) and root knot nematode (*Meloidogyne incognita*) [83]. 

Some nematodes have acquired the ability to recruit plant β-glucanases to the infection sites as a way to promote virulence by assisting in the development of giant cells and syncytia, specialized feeding structures [83,84]. Although the formation of these structures is different depending on the species, it often involves extensive remodelling of both host and nematode walls, allowing for cell elongation and the appearance of elaborate ingrowths aiming to increase the surface area for nutrient uptake [84,85,86]. Evidence suggests that most of these wall modifications arise from the action of plant enzymes (e.g., β-glucanases), rather than enzymes of nematode origin. Expression of GmCel7, a homolog of Arabidopsis Cel2 (GH9B subclass; Figure 2), is highly induced in nematode-infected roots. While suppression of GmCel7 did not seem to modify the plant wall architecture under normal developmental conditions, it reduced the amount of feeding structures and mature *H. glycines* females present in infected roots by 50% [83]. Although its substrate specificity is unknown, these results suggest that GmCel7 is required for the full virulence of *H. glycines*. The specific induction GH9B and GH9C plant β-glucanases during plant–nematode interactions is well documented in soybean, tobacco and Arabidopsis, suggesting that the mechanism might be an extended nematode virulence strategy [83,84,87,88,89].

CBM49-containing β-1,4-glucanases (GH9C) are thought to be involved in the degradation of crystalline cellulose associated with irreversible wall disassembly such as root hair emergence or breakdown of the endosperm wall during germination [90,91,92]. Experimental evidence shows that the CBM49 domain from the tomato β-1,4-glucanase Cel9C binds crystalline cellulose substrates in vitro [92]. Upregulation of the Cel9C transcript has been observed in ripening fruits correlated with irreversible wall degradation. Although the exact physiological role of SlGH9C1 remains elusive, it could function complimentary with GH9B β-1,4-glucanases and other wall-degrading enzymes during the fruit ripening phase.

Structural studies on characterised and putative GH9C class enzymes suggest that CBM49 might be contributing to substrate selection/modification rather than direct hydrolysis of crystalline cellulose [43]. Site-directed mutagenesis highlighted the importance of some aromatic residues in CBM49, contributing to the interaction with the surface of crystalline substrates [92]. Similar residues in homologous CBMs from non-plant GH9 β-1,4-glucanases are important to discriminate between related ligands such as cellulose, xylan, or MLG [93,94,95,96,97]. Although cellulose is the likely physiological substrate for GH9C plant 1,4-glucanases, it cannot be excluded that they are also able to degrade other non-cellulosic polysaccharides in muro [43,91,92]. Supporting this idea, the catalytic domains of SlCel9C1 and its rice ortholog show broad substrate specificity and are able to hydrolyse not only crystalline cellulose but also arabinoxylan or MLG [92,98]. 

In Arabidopsis, AtGH9C1 knock-down mutants show defects in root hair formation and delayed germination. AtGH9C1 is primarily expressed in root hair cells and endosperm cells, two cell types with specialized walls which are fully dismantled during development [91]. It is likely that AtGH9C1 hydrolyses cellulose in these wall types to facilitate wall weakening, although enzyme activity or substrate preference have not been reported. The presence of the CBM49 domain seems to target AtGH9C1 to particular regions of the wall, and when it is missing the protein, it is diffusely distributed throughout the wall [91]. In a different study, it was observed that the CBM domain of a rice GH9C endoglucanase is post-translationally cleaved after the protein was targeted to the wall [98]. The proteolytic cleavage of CBM-containing GH9C proteins might activate their hydrolytic activity or modify their substrate specificity. After activation, GH9Cs could regulate the degree of crystallinity and thus the cross-linking with matrix polysaccharides, controlling cell expansion during development [74]. Downregulation of AtGH9C2 led to decreased cellulose crystallinity associated with an increase in plant height and rosette diameter. Conversely, overexpression of the poplar ortholog PtGH9C2 results in increased cellulose crystallinity and opposite plant phenotypes [74]. Considering the characteristic expression profile of GH9C2 in Arabidopsis and Poplar, it has been proposed that this β-1,4-glucanase controls cell expansion during secondary wall development by regulating the crystallization of cellulose microfibrils and thereby the interaction with other wall polymers [74].

## 3. β-1,3-Glucanases

### 3.1. Classification and Evolutionary Origin

The large diversity of β-1,3-glucanases makes their classification difficult, resulting in a widespread distribution in several clades of the glycoside hydrolase superfamily [99]. Several attempts have been made to classify β-1,3-glucanases in different plant species. Initial classifications were based on sequence similarity, molecular size, or isoelectric point. As a result, a number of distinct β-1,3-glucanases classes have been defined in tobacco [100], barley [8], cotton [101], soybean [102] or rice [103,104]. Despite sharing a significant protein structural similarity and sequence-related attributes, members of the same subgroups showed a high diversity of temporal and tissue-specific expression patterns. For example, 27 rice β-1,3-glucanases were classified into four groups based on amino acid sequence [104]. Enzymes classified within the same group were differentially expressed in various tissues, developmental stages as well as in response to different plant biotic and abiotic stresses and hormone treatments. While OsGlu11 and 27 were specifically expressed in stem and leaf and strongly induced by *Magnaporthe grisea* infection and SA and ABA treatments, several other members of the same family were expressed in flowers and germinating grains [104]. Such classifications allowed for a good separation of β-1,3-glucanases from the sequence-related β-1,3;1,4-glucanases encoded in the same genome [99]. However, the diversity of expression patterns of members of the same group did not allow one to infer common biological roles.

In an attempt to clarify the biological significance of β-1,3-glucanase gene multiplicity and the relationship between sequence and function, more complex classification methods have been established [105,106]. Advanced phylogenetic analysis combined with expression information, knowledge of co-expressed genes, and published data allowed for the grouping of 50 putative β-1,3-glucanases encoded in the Arabidopsis genome in five protein domain architectures (Group I to V; Figure 2). The exon–intron organization of the corresponding genes within these five protein structural archetypes is relatively conserved in distantly related species such as cotton, cacao, and grape vine [106]. The common protein domains shared among these five groups consist of the presence of an N-terminal sequence (NTS) and a core glycosyl hydrolase (GH) domain. Group V is composed by proteins with this basic architecture. The remaining four groups contain additional domains. A carbohydrate-binding module (CBM43) and a hydrophobic C-terminal sequence (CTS) are present after the GH domain in members of group II and IV, respectively. The CBM43 domain is known to allow for the binding to β-1,3-glucans [107], while the CTS sequence encodes a transient transmembrane domain similar to vacuolar targeting peptides and glycosylphosphatidylinisotol (GPI)-anchor attachments [108,109,110]. Group I proteins contain both the CBM43 domain and the CTS region. Finally, group III proteins have a second CBM43 domain, but they do not contain the CTS region (Figure 2).

There seems to be a series of evolutionary events leading to the divergence in β-1,3-glucanase protein domain architectures and expression patterns to predict biological functions [105]. According to this model, ancestral β-1,3-glucanases belonging to the architecture group I show an abundant expression in a variety of tissues and organs and might have a predominant role in wall remodelling during plant development. Additional emerging β-1,3-glucanases (groups II to V) would have appeared during evolution by gene-duplication, subsequent loss of protein domains, and regulatory cis-elements leading to the observed structural and functional diversity. For example, stress-related β-1,3-glucanases would have evolved from group I ancestors with the acquisition of stress-responsive expression patterns followed by loss of the CTS region including the GPI-anchoring site, thus allowing for extracellular secretion upon certain signals. Accordingly, Arabidopsis group II (lacking the CTS domain) is enriched in pathogenesis-related β-1,3-glucanases, whose expression is specifically induced in response to pathogens and thus might be involved in the degradation of β-1,3 glucans present in fungal cell walls [105,106]. 

### 3.2. Biological Roles

Plant β-1,3-glucanases have received significant attention as important players in plant–microbe interactions, as a large number of β-1,3-glucanases are included in pathogenesis-related (PR) group 2 of proteins [24]. This type of β-1,3-glucanases accumulates in the event of a pathogen attack, and some of them have been directly involved in the hydrolysis of pathogen walls, as β-1,3-glucans are found in bacteria, metazoa, viruses, and particularly fungi. In this last group, β-1,3-glucans constitute the most abundant wall structural component existing in the form of linear or branched glucans containing mostly β-1,3 and β-1,6 linkages [111,112,113]. Antifungal hydrolytic activity of β-1,3-glucanases isolated from multiple plant species has been confirmed both in vitro and in vivo. 

Purified PR-2 β-1,3-glucanases often show inhibitory effects on fungal spore germination and on hyphal growth, causing mycelial deformations and lysis due to the hydrolysis of the fungal walls. This type of antifungal activity was shown by the tobacco GluII β-1,3-glucanase against *Fusarium solani* [114]. The antifungal activity of β-1,3-glucanases is often not species specific, and it show inhibitory effects on a wide range of pathogens. For example, extracts from pea plants overexpressing a barley β-1,3-glucanase reduces the spore germination of *Trichoderma harzianum* and *Colletotrichum acutatum* and delays hyphal growth in *Botrytis cinerea* and *Ascochyta pisi* [115]. Similarly, a wheat β-1,3-glucanase heterologously produced shows inhibitory effects on hyphal growth, spore formation and mycelial morphology of *Fusarium*, *Alternaria* and *Penicillium* species [116]. Furthermore, some β-1,3-glucanases have shown antifungal activity against pathogens specialized in infecting distantly related plant clades. For example, wheat TaGluD showed in vitro antifungal activity not only against monocot-specific *Rhizoctonia*
*pathogenic* strains, but also against *Phytophthora capsici* and *Alternaria longipes*—specialized pathogens infecting dicot hosts such as hot pepper and tobacco, respectively [117]. 

Direct antifungal effects have also been observed in vivo. For example, functional studies of rice–*Magnaporthe oryzae* interactions demonstrated a direct antifungal effect of the Gns6 β-1,3-glucanase dependent on its hydrolytic activity [118]. While wildtype Gns6 was able to inhibit conidial germination and formation of appressoria, a non-catalytic gns6 mutant lost all antifungal effects, showing enhanced pathogen colonization and increased disease susceptibility [118]. Furthermore, differential expression of PR-2 β-1,3-glucanases has been associated with increased disease resistance when comparing susceptible and resistant cultivars in a number of crop species such as rocket salad, tomato, and wheat [119,120,121]. Transgenic expression of β-1,3-glucanase genes has been extensively used as a strategy to develop durable disease resistance against fungal pathogens in crop plants [24]. For example, overexpressing a tobacco β-1,3-glucanase in transgenic groundnut plants significantly improved disease resistance against the pathogens *Cercospora arachidicola* or *Aspergillus flavus*, reducing the number and size of the necrotic lesions and a general delay in the disease symptom progression or preventing aflatoxin accumulation in seeds, respectively [122]. Similar protective effects have been reported by overexpressing potato, barley, or soybean β-1,3-glucanases in flax, wheat, and banana transgenic lines [123,124,125]. In general, these PR-2 β-1,3-glucanases belong to the Group V, characterized by the presence of a secretion signal and the absence of CBM and CTS domains (Figure 2).

Plant β-1,3-glucanases are also involved in an alternative plant defence mechanism, where degradation of the hyphal cell wall of invading fungi would release β-1,3-glucan oligosaccharides serving as elicitors. Upon recognition by plant surveillance systems, these β-1,3-glucan oligosaccharides would act as microbe-associated molecular patterns (MAMPs) triggering the activation of signalling cascades resulting in a wide range of localized and systemic defence responses [126]. For example, upon oomycete infection, soybean and rice β-1,3-glucanases are able to release β-glucan elicitors from the fungal wall, which trigger the production of antimicrobial compounds such as phytoalexins [127,128,129]. Likewise, a synthetic β-1,3/-1,6-glucan heptaglucoside similar to those obtained after β-1,3-glucanase hydrolysis from fungal walls is able to activate plant defences in legume species [130,131]. The exact mechanism of recognition and mode of action of pathogen-derived β-glucan elicitors is not fully understood, and it seems to be dependent not only on the plant species and the origin of the β-glucan elicitor, but also on the glycan structure, where length, branching pattern and presence of chemical modifications have shown to be relevant [132]. Recent evidence suggests that this mechanism might be more widespread among the plant kingdom than expected, and diverse β-1,3 glucanase-derived oligosaccharides were recently demonstrated to be potent elicitors of plant defences in *Nicotiana benthamiana*, barley, rice, and *Arabidopsis* [133,134]. 

In addition to the generation of β-glucan elicitors, recent evidence suggests that plant β-1,3-glucanases are also involved in the release of soluble β-glucans from fungal walls with antioxidant activities required not only for pathogenicity, but also for the successful establishment of root interactions with beneficial endophytic fungi [135]. Barley HvBGLUII, an apoplastic Group V β-1,3-glucanase, is able to partly hydrolyse Serendipita indica walls. As a result, a non-immunogenic β-1,3;1,6-glucan decasaccharide (β-GD) with ROS scavenging properties is released, thus facilitating fungal development. Exogenous applications of β-GD increase the fungal root colonization efficiency. In addition, these effects are highly specific to the length and branching pattern of the oligosaccharide, as altering the glucan structure blocks its activity completely. It has been proposed that such a hijacking of plant apoplastic β-1,3-glucanases might be a common fungal counter-defensive strategy to subvert host immunity during pathogen and beneficial fungi colonization [135]. Given the heterogeneity of 1,3- and 1,3-1,6-glucan structures found in fungi and other microbes, the number of biological processes that involve such glucan effectors resulting from the hydrolytic action of plant β-glucanases might be larger than previously considered.

In addition to their roles during plant–microbe interactions β-1,3-glucanases also play a role in the degradation of plant produced callose, a transient β-1,3-glucan polymer accumulated at discrete sites during certain developmental processes and in response to environmental cues. For example, during microsporogenesis, another specific β-1,3-glucanases degrades the thick callose wall surrounding the tetrad, releasing the microspores into the anther locule for pollen maturation [136,137,138]. Rice mutants harbouring a catalytically inactive Osg1 β-1,3-glucanase produce damaged pollen grains. In those mutant pollen grains, large amounts of callose remains, and microspores remain attached to each other resulting in deformations and male sterility [139]. In a less understood way, stigma- and style-specific extracellular β-1,3-glucanases have been involved in pollination, specifically in the degradation of callose in the stylar matrix during pollen tube growth [140]. 

Another example of the involvement of β-1,3-glucanases in plant development is the control of callose degradation in plasmodesmata (Pd). Pd form channels physically interconnecting the cytoplasm and endoplasmic reticulum of adjacent cells [141,142]. β-1,3-glucanase-mediated callose degradation is thought to be tightly regulated in order to control the symplastic trafficking of micro- and macromolecules, including phytohormones, mRNA, and mobile non-cell-autonomous transcriptional factors [143,144]. Controlled build-up and degradation of callose deposits in Pd is relevant for multiple developmental processes such as germination, axillary bud growth, lateral root organogenesis, shoot apical meristem determination, or stomatal differentiation [145,146,147,148,149]. In addition, degradation of callose in Pd has been observed in response to environmental signals such as low temperatures, circadian rhythm or pathogen attack [141]. This type of Pd-specific β-1,3-glucanase contains a GPI-anchor targeting them to the plasma membrane. Moreover, they can also have a CBM43 domain, but its presence is facultative. Hence, they are classified into groups I or IV (Figure 2). They show a characteristic punctuate pattern around Pd colocalizing with callose deposits. In Arabidopsis, Pd-localized β-1,3-glucanase knockout mutants contain larger callose deposits, causing a decrease in Pd conductivity and cell-to-cell transport defects [143,146]. In perennial plants such as birch (*Betula pubescens*) or tree peonies (*Paeonia suffruticosa*), breakage of bud dormancy after the winter season is mediated by a Pd-specific β-1,3-glucanase in charge of degrading callose deposits and thus restoring symplasmic organization [150,151]. Two CBM43-containing enzymes—PsBG6 and PsBG9—have been implicated in this process. Despite both of them having been Pd-localized, PsBG6 was found inside of the Pd passage, whereas PsBG9 was observed to enclose the Pd channel, correlated with the presence of a GPI-anchor only in PsBG6 (Group I) and not in PsBG9 (Group II) (Figure 2; [151]).

In tobacco, a similar β-1,3-glucanase, ßGluI, regulates the release of coat-imposed dormancy and germination by promoting the rupture of the testa [152]. In tomato and banana, the expression of GPI-anchored plasmodesmata-located β-1,3-glucanases has been linked to fruit formation and softening through the modulation of symplasmic unloading of sugars and signalling molecules into the ovaries and fruit tissues [153,154].

Pd cell-to-cell communication channels are also utilized by viral pathogens to move and infect their plant hosts. β-1,3-glucanase mutants affected in the callose degradation in Pd show increased viral particle movement and enhanced susceptibility [155,156,157]. Several virus, bacteria and filamentous pathogen effectors have been shown to target Pd conductivity [158]. Potato virus Y is able to induce plant Pd-localized β-1,3-glucanase activity as an infection strategy resulting in the degradation of callose in Pd increasing viral movement and spread [159]. This β-1,3-glucanase upregulation and callose degradation is not observed in resistant potato cultivars where the virus fails to spread [160]. 

Callose is also accumulated in the form of cell wall thickenings at the site of contact with pathogens such as viruses, fungi or bacteria, and in wounds caused by herbivory attack. As part of the complex transcriptional reprograming regulated by the absiscic acid- jasmonic acid hormonal crosstalk, specific β-1,3 glucanases are downregulated during incompatible plant virus interactions [161,162]. Similarly, β-1,3-glucanase downregulation is observed during foliar fungal infections, where haustorial feeding structures are rapidly surrounded by callose deposits [163]. However, biochemical characterization of the β-1,3-glucanases involved in this type of defence-related callose deposition is scarce.

In winter cereals, the expression of Group V β-1,3-glucanases with anti-freezing activity has been associated with increased chilling tolerance [164]. For example, in rye (*Secale cereale* L.) freezing temperatures induce the expression of ScGLU-2 and ScGLU-3 β-1,3-glucanases. In vitro, these β-1,3-glucanases partially retain hydrolytic activity at sub-zero temperatures and have the ability to limit the growth of ice crystals. Structural modelling identified ice-binding surfaces (IBS) specifically in these β-1,3-glucanases, geometrically complementary to the surface of ice crystals. Residues on the putative IBSs are charge conserved in tolerant varieties but not in similar β-1,3-glucanases from non-acclimated rye varieties [165]. Similarly, accumulation of the VcGNS1 β-1,3-glucanase was observed in the skin of harvested table grapes after a prolonged cold treatment. Recombinant VcGNS1 is stable at 0 °C and is able to delay the inactivation of some enzymes after repeated freeze–thawing cycles [166]. Although it is not clear whether glucan hydrolytic activity is required for these effects, expression of β-1,3-glucanases with in vitro cryoprotectant activity has been correlated with chilling tolerance in species such as tobacco, tomato, and spinach [166,167,168].

## 4. β-1,3-1,4-Glucanases

Phylogenetic evidence supports the hypothesis that plant β-1,3-1,4-glucanases evolved from an ancestral Group V β-1,3-glucanase from the widely distributed family GH17 (Figure 2). The accumulation of only a limited number of point mutations during evolution, causing minor differences in the amino acid sequence at the substrate-binding and catalytic sites, was sufficient to develop this unique substrate specificity [8,32,169]. Thus, predictions of β-1,3-1,4-glucanase or β-1,3-glucanase activities based only on amino acid sequence information are often unsuccessful, and biochemical characterizations are required.

A handful of β-1,3-1,4-glucanases from monocot species have been characterized [170,171,172,173,174,175,176]. Often, cereal species encode several β-1,3-1,4-glucanase isoforms with different temporal and tissue-specific expression patterns. In cereals with an MLG-rich endosperm such as barley or rice seed-specific β-1,3-1,4-glucanase, expression is detected rapidly after seed imbibition and reaches a maximum at early stages during germination. In most cases, a second isoform is expressed in the grain endosperm during germination, but it is also detected during seedling elongation and in vegetative tissues such as leaves and roots. For example, two barley β-1,3-1,4-glucanases EI and EII have been characterized, showing high sequence similarity. While expression of EII is restricted to the scutellum during early gemination stages, EI expression is also detected in adult roots and leaves [170,177,178,179,180]. Similarly, rice EGL1 and EGL2 isoforms show a different spatiotemporal pattern [174] EGL2 being seed-specific and EGL1 expressed also in vegetative tissues, reaching maximum values at full expansion and then decreasing upon leaf aging. 

### Biological Roles

Since expression of seed-specific β-1,3-1,4-glucanases is restricted to embryonic tissue, their function is likely to loosen the walls during the course of endosperm mobilization. During germination, breaking down this physical barrier is required to allow for access of additional hydrolytic enzymes to substrates within the cell, such as proteins or starch [170,181].

The functional role of β-1,3-1,4-glucanases in developing leaves and roots is less clear, and several hypotheses have been proposed. The asymmetrical conformation of MLG due to the irregular distribution of β-1,3-linkages limits the capacity of the polymer to aggregate into fibrillar structures such as cellulose. Instead, MLG forms a gel-like matrix able to closely associate with cellulose and xylan to reinforce the wall and guide the orientation of cellulosic microfibrils during growth [182,183,184]. As the developmental pattern of characterized β-1,3-1,4-glucanases in barley, rice, or wheat seemed to indicate that the enzymes accumulate in vegetative organs during the growing phase, it was hypothesized that β-1,3-1,4-glucanases could play a role in loosening the wall by partially hydrolysing MLG, allowing for turgor pressure-driven cell expansion [169,185,186]. However, several pieces of evidence do not support this hypothesis. An increase in the expression of EI is observed after transferring barley seedlings from a normal light/dark photoperiod to continuous darkness. Despite a detection of concomitant decrease in MLG abundance, there was no measurable elongation of leaves, suggesting that wall loosening does not occur [187]. Additionally, the dark-induced accumulation of EI and MLG degradation in leaves is strongly inhibited by high glucose availability in the growth media, indicating that β-1,3-1,4-glucanase activation might be uncoupled from cell expansion. Instead, it has been postulated that MLG could be used by plants as a short-term glucose supply and thus metabolizable energy stored in the wall [187]. According to this second hypothesis during dark cycles, low glucose levels would reach a certain threshold, triggering the induction of β-1,3-1,4-glucanase activity. As a consequence, MLG would be partially depolymerized, and the resulting oligosaccharides could be easily converted into glucose by accessory proteins. The released glucose would be immediately available within the plant providing a flexible energy source under sugar-depleting conditions. This relatively simple catabolic process would represent an advantage in the utilization of MLG as a carbon source over starch and other wall polysaccharides [181]. However, the recent characterization of the first β-1,3-1,4-glucanase mutants in maize and brachypodium raised some reservations regarding this hypothesis and revealed new aspects of MLG degradation in cereals [175,176]. In maize, MLG degradation in vegetative tissues is dependent on a single β-1,3-1,4-glucanase. Mixed Linkage Glucan Hydrolase 1 (MLGH1) has a high sequence homology and expression pattern compared to barley EI or rice EGL1. Although the maize genome encodes several genes annotated as putative β-1,3-1,4-glucanases, the sole disruption of MLGH1 blocks all detectable dark-induced MLG degradation in mutant seedlings [175]. An extended time course analysis showed that, similar to starch accumulation of MLG in maize, seedlings follows a circadian rhythm with the maximum MLG abundance observed at the end of the day and constant decrease once the dark period starts. Transcript accumulation of MLGH1 correlates with the degradation of MLG during these cycles. While mutant plants are not able to degrade MLG and no cycling is observed, MLGH1 overexpression avoids MLG accumulation, and only trace amounts of MLG can be detected [175]. Similarly, the single disruption of the brachypodium homolog, LCH1, results in loss of all detectable dark-induced MLG degradation in vegetative tissues [176]. These results indicate that there is no or limited functional redundancy among β-1,3-1,4-glucanases, and the existence of different isoforms encoded in the genome of individual species might respond to the need of stress- or spatiotemporal-specific expression patterns. Despite the impediment in MLG degradation maize and brachypodium β-1,3-1,4-glucanase, mutants do not exhibit significant developmental or growth-related defects showing similar plant height, vegetative biomass, and grain yield as wild-type plants, suggesting that the hydrolysis of MLG as a source or energy during the night seems not to be essential under normal growth conditions [175,176]. A possible explanation is that under these conditions, starch may serve as the major energy storage polymer, thus allowing mutant plants to maintain their growth rate despite being unable to utilize MLG as a carbon source. However, no significant differences were observed in starch degradation in dark-treated maize mlgh1 seedlings compared to the wild-type, and only a slightly faster starch turnover was reported in adult brachypodium lch1 mutant plants. One possibility is that MLG turnover makes a difference only in specific cell types or under specific circumstances such as plants growing under biotic/abiotic stresses. Hence, these β-1,3-1,4-glucanase mutants are valuable tools to address the biological significance of MLG in cereals in the future.

The implication of diverse plant β-1,3-1,4-glucanases in stress responses such as plant–pathogen interactions has long been proposed, and new evidence has emerged. The expression of some β-1,3-1,4-glucanase genes in rice is induced by wounding, infection with virulent strains of the blast fungus (*Magnaporthe grisea*), and treatment with salicylic acid or methyl jasmonate defence-related phytohormones [103,174,188]. Not surprisingly, rice plants overexpressing the GNS1 β-1,3-1,4-glucanase show an increased disease resistance to *M. grisea*. These mutant plants form spontaneous brown specks accompanied by constitutive activation of defence-related genes and stunted growth [174]. This lesion-mimic phenotype suggests that constitutive degradation of MLG could somehow trigger a signalling pathway leading to constitutive activation of plant defences. Recent reports demonstrate that β-1,3-1,4-glucanase-derived oligosaccharides are recognized by plants and act as defence signalling molecules. Even exogenous applications of short MLG fragments have a protective effect against pathogen attack not only in monocot, but also dicot plant species [189,190]. Dicot plant walls do not contain MLG, raising the question of why these plants can recognize products of MLG degradation. Although the mechanism has not been elucidated, it has been suggested that dicot plants could use β-glucanases to partially degrade MLG present in the walls of pathogens. MLG hydrolysis might directly hinder the pathogen growth, or alternatively, the resulting (1,3;1,4)-β-d-glucooligosaccharides could act as pathogen-associated molecular patterns (PAMPs), whose detection would trigger the activation of downstream plant defences. The presence of MLG in plant pathogens is poorly characterized and has only been demonstrated in a few examples, such as in the fungi *Rhynchosporium secalis*, *Aspergillus fumigatus* and *Neurospora crassa*, the oomycete *Hyaloperonospera arabidopsidis* or the endosymbiotic bacterium *Ensifer meliloti* [190,191,192,193,194]. MLG oligosaccharide-triggered protection has been reported for pathogens with completely different lifestyles and infection strategies such as the necrotrophic pathogens *Sclerotinia sclerotiorum* and *Botrytis cinerea* or biotrophs such as the oomycete *Hyaloperonospora arabidopsidis* or the hemibiotrophic bacterium *Pseudomonas syringae*. Considering these results, it has been speculated that MLG might be a more widespread wall component in plant pathogens than previously thought [189,190]. The existence of such a mechanism could imply the presence of hitherto uncharacterized plant β-glucanases encoded in the genome of dicot species that generate MLG-derived signalling molecules [26,189]. Thus far, no β-1,3-1,4-glucanase has been identified in *Arabidopsis*. Alternatively, β-1,4-glucanases might be responsible for the production of immunogenic MLG oligos, as these enzymes hydrolyse internal β-(1,4)-glucosidic linkages in MLG, although the products of hydrolysis are different from those derived from β-1,3-1,4-glucanase action (Figure 1) [8,195]. Supporting that hypothesis, MLG oligosaccharide structures compatible with cellulase origin display similar defence-related responses than those derived from β-1,3-1,4-glucanase hydrolysis [190]. Unfortunately, only a small number of the β-glucanases from *Arabidopsis* or other dicot species have been biochemically characterized. Hence, the study of such enzymes represents an exciting task for the future.

## 5. Conclusions

Plant β-glucanases catalyse the hydrolysis of β-glucosidic linkages found in the structure of polysaccharides present in the cell wall of plants and microbes (Figure 1). This catalytic ability confers various physiological roles to plant growth, development and interaction with the environment (Table 1). Despite recent advances in identifying new physiological functions, characterization of the precise biochemical activities is needed to decipher the importance of β-glucanases in plant adaptation towards environmental cues.

## Figures and Tables

**Figure 1 plants-11-01119-f001:**
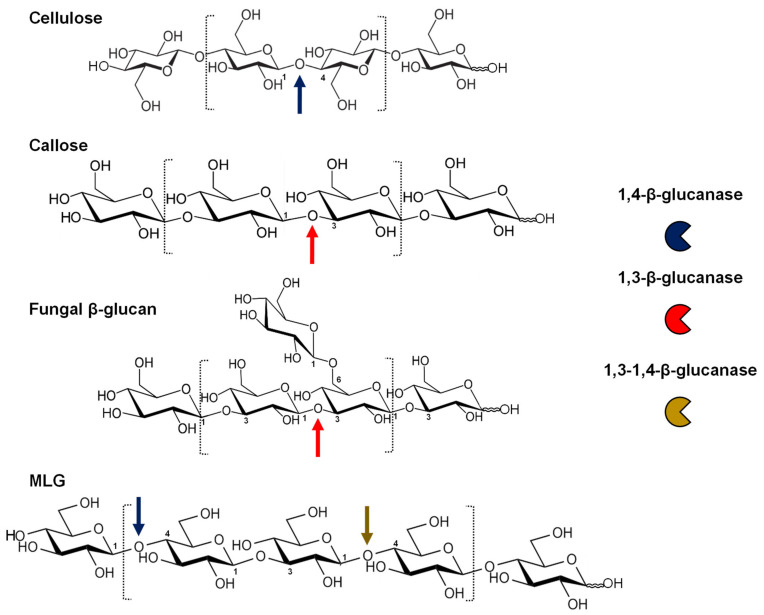
Activity of the three types of β-glucanases found in plants and their main physiological substrates. (1,3)- and (1,4)-β-glycosidic linkages are depicted as 3 and 4, respectively. Arrows indicate the glycosidic linkage hydrolysed in each case.

**Figure 2 plants-11-01119-f002:**
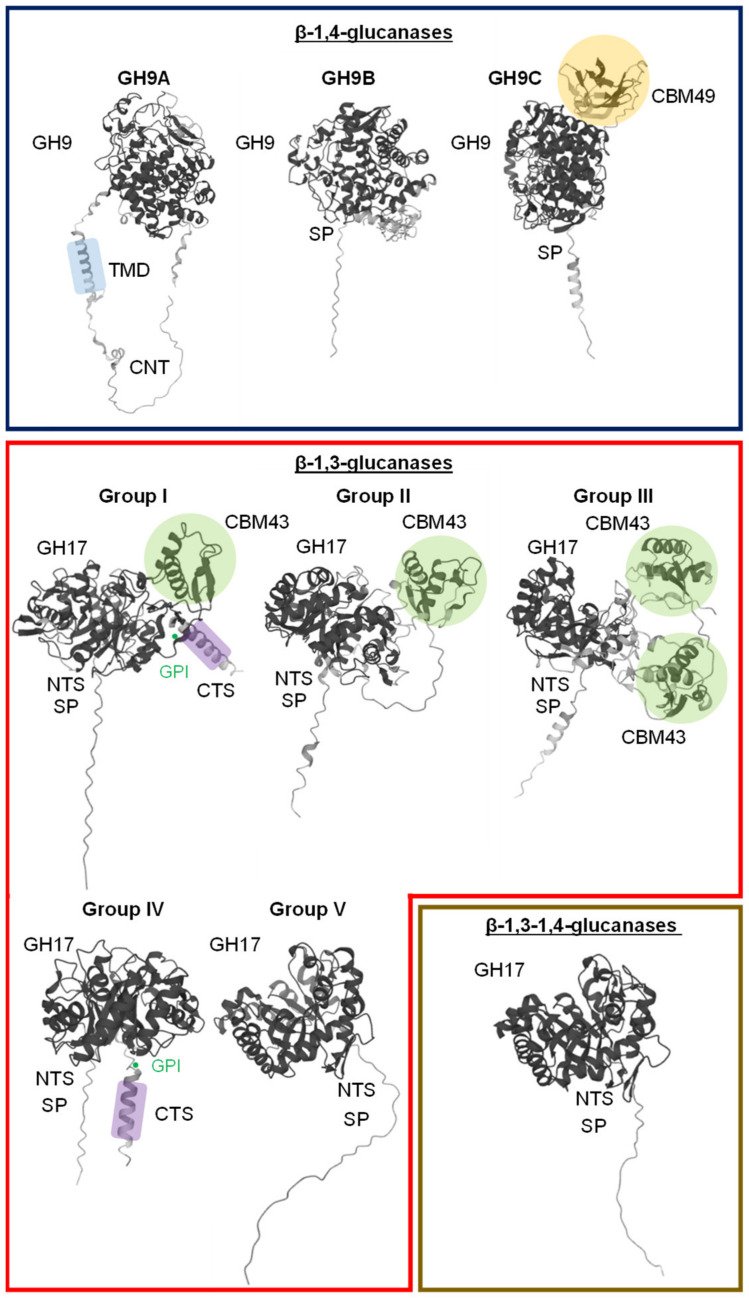
Topology models of plant β-glucanases. Protein domains and regions are abbreviated as follows: NTS—N-terminal sequence; GH—core glycosyl hydrolase family domain; CBM—carbohydrate binding module; GPI—glycosylphosphatidylinisotol-anchor attachment; CTS—hydrophobic C-terminal sequence; CNT—cytosolic N-terminal extension; SP—secretory signal peptide; TMD—transmembrane domain. Structure models represented in this figure were built by using the AlphaFold Protein Structure Database [40,41] with the following Uniprot accessions: GH9A 1,4-β-glucanase (AtGH9A1; Q38890), GH9B 1,4-β-glucanase (AtGH9B16; Q9SVJ4) and GH9C 1,4-β-glucanase (AtGH9C1; Q9M995), group I 1,3-β-glucanase (At5g64790; Q9LV98), group II 1,3-β-glucanase (At2g05790; F4IHD3), group III 1,3-β-glucanase (At2g39640; O48812), group IV 1,3-β-glucanase (At1g77780; Q9CA16), group V 1,3-β-glucanase (At5g20340; O49353), 1,3;1,4-β-glucanase (ZmMLGH1; B6T391).

**Table 1 plants-11-01119-t001:** Proposed physiological roles of plant β-glucanases.

Type	Substrate	Physiological Roles
1,4-β-glucanases	Crystalline cellulose	Irreversible wall disassembly: root hair emergence, endosperm breakdown, fruit ripening
	Amorphous cellulose	Secondary wall mechanical strength Wall remodelling: cell expansion, fruit ripening, nematode attack Cell-cell adhesion: grafting, plant parasitism Cellulose biosynthesis (plasma membrane-associated)
	MLG ^1^ (Fungal wall)	Antifungal activity Elicitor release: MAMP ^2^
1,3-β-glucanases	Callose	Plasmodesmata and symplastic transport: dormancy release, fruit development, cell-to-cell communication Reproductive organs: pollen, style and stigma development
	1,3-β-glucan (Fungal wall)	Antifungal activity Elicitor release Fungal effector release
1,3-1,4-β-glucanases	MLG ^1^	Cell wall loosening during germination Energy source in the dark
	MLG ^1^ (Fungal wall)	Antifungal activity Elicitor release

^1^ MLG—mixed-linkage glucan; ^2^ MAMP—microbe-associated molecular patterns.

## Data Availability

Not applicable.

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
