# Peer review of "Emerging Roles of β-Glucanases in Plant Development and Adaptative Responses"

_plants, 2022, doi:10.3390/plants11091119_

Round 1

Reviewer 1 Report

Dear Authors,

I have  great honor to assess the review manuscript entitled: “Emerging roles of β-glucanases in plant adaptative responses” which is considered for publication in Plants journal. The review manuscript is generally good written but in my opinion need some improvements in the case of extend of topic present and Figure presentation. The specific comments I present below in a form of list specific comments :

Minor suggestions

I suggest to move Figure 1 to the next page and enlarge it as big as possible. Because, now the Figure has easily to see bad pixels. Moreover, this figure has information about of β-glucanases activity then in my opinion particular activities should be marked on Figure 1.

Figure 2 I strongly recommend to divide this Figure to two separate because now the elements of 3D structures is difficult to read. After separation of Figures enlarge structures as big as it is possible

Major suggestions:

I recommend to add the more information about β-glucanases and other cell wall associated changes in context of adaptation or resistance to plant viruses. This fact was widly investigated since 2018 also in MDPI journals like IJMS but others. This elements will make this review more complete and more interesting. Because of amount of time needed for add new information I recommend major revision but generally this is good piece of scientific work.

Sincerely,

Author Response

Please see the attachment (word document named Responses to reviewer 1).

Reviewer 2 Report

The review is well written, the three categories of plant β-glucanases have been comprehensively described. This piece of work is very informative and helpful to a wide audience. However, I would like to mention a few points which might help the readers to navigate through this work and understand easily. 

  1. The biological roles can be categorized into plant growth and development, plant physiology, plant-pathogen/herbivore interactions (symbiotic, parasitic, etc.), etc., with sub-headings.
  2. A more clear description of how/whether they are involved in PTI or ETI response.
  3. More elaborate discussion and conclusion on how these information can be used to advance further research and crop improvement. 
  4. A comprehensive schematic diagram of the various roles of plant β-glucanases will be very helpful to the readers. 
  5. The title needs to be more general/modified as the information inside the review is much more than only the adaptative responses. 

Author Response

Please see the attachment (word document named Responses to reviewer 2).

Round 2

Reviewer 1 Report

Dear Authors,

All my suggestions was added to the manuscript I recomend publication

Sincerely,

Reviewer 2 Report

I could not figure out the amended “conclusion” section (lines 739-741), as mentioned by the authors.